# MAIG: Multi-agent system for Academic Illustration Generation based on deep search and reflection

## Abstract

While text-to-image models have revolutionized creative content generation, they fall short in the domain of academic illustration, which demands stringent scientific accuracy and informational completeness, creating a significant bottleneck in automated scientific communication. Existing models often produce illustrations that are factually incorrect, omit critical information, and are limited to simple structured diagrams, failing to render the complex, unstructured conceptual visuals common in science. To address these challenges, we introduce **MAIG**, a novel multi-agent framework that mimics an expert's workflow. MAIG first employs a deep research agent to ground the generation process in a factual knowledge base, ensuring all necessary background information is available. Subsequently, reflection and editing agents iteratively verify the visual output against this knowledge, identifying and correcting scientific errors. In the meantime, evaluating scientific figures is a parallel challenge plagued by subjective and unscalable methods, we also propose a novel Question-Answering (QA) based Evaluator. This method leverages the strong reasoning capabilities of modern Multimodal Large Language Models (MLLMs) to quantitatively measure both informational completeness and factual correctness, providing an objective and scalable assessment of an illustration's quality. Extensive experiments across various scientific disciplines demonstrate the effectiveness of MAIG, which achieves minimal factual errors and the most complete knowledge coverage, significantly outperforming state-of-the-art models. Our results validate that the proposed research-reflect-edit loop is crucial for generating high-fidelity scientific illustrations and that our QA-based evaluator offers a reliable assessment methodology, together forming a comprehensive solution for advancing automated scientific visualization.

## 1 Introduction

Recent advances in text-to-image synthesis have achieved remarkable success in artistic and photorealistic generation. However, this progress has not translated to domains where precision is paramount, such as academic illustration. Unlike creative works, scientific figures are governed by a ground truth and must satisfy stringent requirements: (1) Informational Completeness, ensuring all intended concepts are fully expressed; (2) Factual Correctness, representing scientific knowledge without ambiguity or error; and (3) Conceptual Versatility, handling both simple structured diagrams and complex unstructured visuals.

Despite some prior efforts, existing generative models consistently fail to meet these demands. They suffer from critical failures: (1) They produce informationally incomplete figures, either by omitting details from the prompt or by failing to incorporate essential background knowledge. (2) They are plagued by factual inaccuracies, misrepresenting key concepts even when explicitly described. (3) Their capabilities are limited to generating simple, structured charts, leaving the complex unstructured visuals such as biology conceptual diagrams unaddressed.

To overcome these limitations, we introduce MAIG (as shown in Fig.1), a Multi-agent system for Academic Illustration Generation, designed to directly tackle these core failures: (1) To ensure informational completeness, it employs a deep research agent that actively searches for and integrates

necessary background knowledge. (2) To guarantee factual correctness, it incorporates reflection and editing agents that critically evaluate the generated image and iteratively correct scientific errors. (3) For efficiency, a task routing module intelligently bypasses these intensive steps for simple drawing requests.

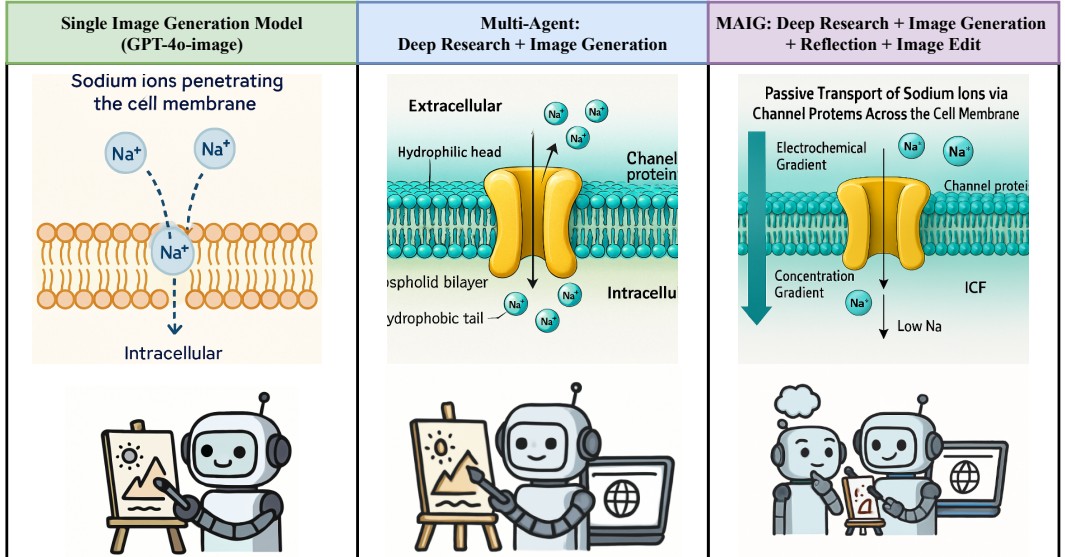

Figure 1: Our proposed multi-agent system MAIG integrates image generation models with deep research modules, reflection and editing modules. Compared with single image generation models and multi-agent systems that only use deep research, it effectively improves the scientific and information completeness of academic illustration drawing.

Beyond generation, a parallel challenge exists in evaluation. Assessing the scientific quality of an illustration is a samely difficult task, and existing methods are inadequate: (1) Irrelevant Automatic Metrics: Standard metrics like FID or CLIP Score measure visual style but are completely insensitive to scientific correctness. (2) Subjective and Slow Manual Evaluation: Relying on human experts is costly, time-consuming, and suffers from inter-rater variability, hindering rapid and scalable benchmarking.

Therefore, we propose a novel Question-Answering (QA) based Evaluator, a scoring mechanism that leverages the strong reasoning capabilities of modern Multimodal Large Language Models (MLLMs) for objective and scalable evaluation. By generating targeted questions that probe for specific scientific facts within an illustration, our method can quantitatively measure both informational completeness and factual correctness.

To validate our contributions, we conduct comprehensive experiments. The results demonstrate that MAIG significantly outperforms state-of-the-art generative models across multiple scientific disciplines. Furthermore, we show that our QA-based evaluation paradigm provides a reliable and objective measure of illustration quality. Our main contributions are summarized as follows:

(1) We propose MAIG, a novel multi-agent framework that operationalizes a research-reflect-edit loop to overcome the core limitations of standard T2I models for scientific tasks.

(2) We introduce a QA-based Evaluator, providing the first objective, scalable, and scientifically-grounded method for assessing academic illustrations.

(3) We perform extensive experiments that validate the effectiveness of our generation framework and the reliability of our evaluation method.

## 2 RELATED WORKS

### 2.1 DRAWING ILLUSTRATIONS FOR ACADEMIC PAPERS

Recent work has moved from pixel-level drawing to structure-aware, editable generation. *DiagrammerGPT* and *From Words to Structured Visuals* formalize text-to-diagram with planning/agent pipelines and accompanying benchmarks, emphasizing structural coherence and editability Zala et al.; Wei et al. (2025). *Draw with Thought* further targets editability by reconstructing scientific diagrams into executable mxGraph/XML via training-free multimodal reasoning Cui et al. (2025). For native vector graphics, *AutomaTikZ* learns to synthesize TikZ code directly from text, yielding publication-ready figures Belouadi et al.. Foundational corpora such as *AI2D* and *AI2D-RST* provide densely annotated scientific diagrams that underpin structure-aware understanding and generation Hiippala & Orekhova (2018); Hiippala et al. (2021). For result figures, chart-centric efforts include *AskChart* for universal chart understanding and $C^2$ for scalable auto-feedback in LLM-based chart generation, while *CharXiv* offers a realistic evaluation suite for charts appearing in papers Yang et al. (2024); Koh et al. (2024); Wang et al. (2024). Beyond figures, *paper2poster* automates scientific communication end-to-end by converting papers into editable posters via parsing, planning, and rendering loops Pang et al. (2025). Overall, these trends point toward semantically grounded, program- or graph-based outputs that better support scholarly authoring and post-editing.

### 2.2 MULTI-AGENT IMAGE GENERATION SYSTEM

Recent work frames text-to-image (T2I) creation as a coordinated pipeline of specialized agents. Proactive, multi-turn systems add uncertainty-aware clarification and belief-graph editing to improve intent alignment Hahn et al. (2024); Chen et al. (2025), while training-free planning frameworks decompose complex instructions into executable steps Liu et al. (2025c). For compositional scenes, agentic scene parsing and region-wise refinement enhance object–relation fidelity Li et al. (2025). Interactive editing extends this paradigm to dialogue-driven adjustments, affect-conditioned manipulation, and layout-preserving, reward-guided revisions Qiu et al. (2025e;b;d). Beyond single-image tasks, collaborative "creator–critic–enhancer" teams and workflow agents automate real production graphs, from benchmarking agent design in ComfyUI to self-optimizing workflow synthesis Crea et al. (2025); Liu et al. (2025a;b). Domain-targeted systems further broaden scope, including culturally aware multi-agent generation, safety-critical synthetic data creation, and urban street-design pipelines that integrate localization, prompt optimization, generation, and automated evaluation Qiu et al. (2025c;a); Liu et al. (2025d). Overall, these multi-agent systems operationalize a parse–generate–evaluate loop that increases controllability, editability, and robustness for image generation in practical settings.

## 3 ILLUSTRATION GENERATION AND EVALUATION

This section details the methodology for generating and evaluating academic illustrations using our MAIG framework. We define two primary tasks—context-rich paper illustration generation and context-scarce textbook illustration generation—and address their unique challenges by employing two distinct, specialized pipelines within a multi-agent system. To assess the results, we introduce a novel "question-answer" evaluation protocol, which utilizes a multi-agent collaboration strategy to objectively measure the scientific quality of the generated outputs.

### 3.1 TASK ROUTING

As depicted in Fig. 2, a Task Routing module, implemented using a Large Language Model (LLM), first analyzes the user's query to classify it into one of two distinct tasks. This initial classification is crucial as different multi-agent strategies are designed to handle the unique characteristics of each request type:

- **Paper Illustration Generation**: This task focuses on visualizing novel knowledge, concepts, and discoveries for research publications. These inputs are characterized by the presence of rich contextual information, specifically a Caption and a corresponding text Section, which serve as detailed prompts for the generation model.

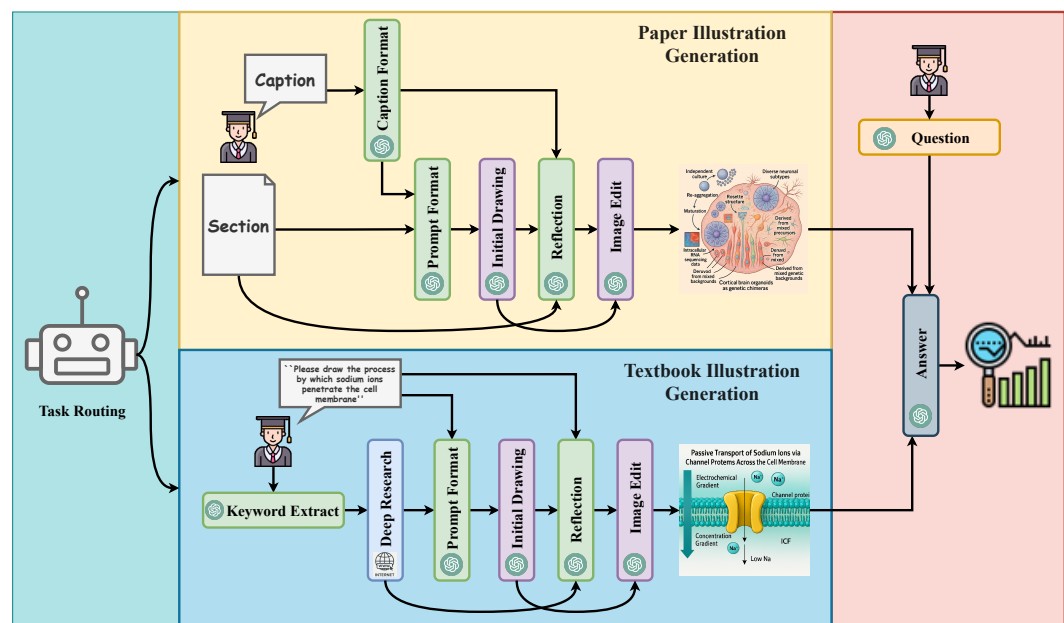

Figure 2: The framework of our multi-agent system of academic illustration generation. After the task routing determines the task type, two different strategies are used to handle two different tasks. Through modules such as prompt word reconstruction, deep network search, reflection, and secondary editing, the generated results of multi-agent systems are more scientific and comprehensive.

- **Textbook Illustration Generation**: This task is centered on creating visuals for established, common-sense knowledge used in teaching. These inputs are typically concise, self-contained requests (e.g., "Draw the process of sodium ions passing through the cell membrane") that lack extensive descriptive text and often require the system to retrieve background knowledge.

Once classified, the input is dispatched to the appropriate specialized pipeline for generation.

## 3.2 PAPER ILLUSTRATION GENERATION

For the task of generating paper illustrations, the first thing we need to do is to use LLM to organize $Caption$ and $Section$ into prompts that can be used by image generation models. Specifically, we will transform the $Caption$ into a descriptive statement $requirement$ that can be visualized, and then further expand it by incorporating supplementary knowledge from the $Section$

$$requirement = \textbf{Caption Format}(Caption)$$

$$formatted\ prompt = \textbf{Prompt Format}(requirement, Section)$$

We call the image generation model to generate the initial image $init\ image$ based on the prompt words $formatted\ prompt$:

$$init\ image = \textbf{Initial Drawing}(formatted\ prompt)$$

By utilizing the powerful understanding ability of MLLM, we compare the image generation requirements and reference knowledge based on $init\ image$, and generate reflective suggestions for secondary modification. The image editing model will make final edits based on reflective suggestions.

$$reflection\ advice = \textbf{Reflection}(requirement, Section, init\ image)$$

$$edited\ image = \textbf{Image\ Edit}(init\ image, reflection\ advice)$$

### 3.3 TEXTBOOK ILLUSTRATION GENERATION

Unlike the generation of paper illustrations, the use of textbook illustrations is often just a simple request from teachers or students in teaching scenarios, lacking background knowledge and information. Therefore, we need to first extract keywords and conduct a deep search for background knowledge. Based on the user's input $requirement$, we use LLM to extract scientific keywords and conduct online deep searches. Specifically:

$$keywords = \textbf{Keyword\ Extract}(requirement)$$

$$ref\ knowledge = \textbf{Deep\ Research}(keywords)$$

Next, we call LLM, the image generation model, MLLM, and the image editing model is used to perform the same generation, reflection, and editing process as the paper illustration generation strategy.

$$formatted\ prompt = \textbf{Prompt\ Format}(requirement, ref\ knowledge)$$

$$init\ image = \textbf{Initial\ Drawing}(formatted\ prompt)$$

$$reflection\ advice = \textbf{Reflection}(requirement, ref\ knowledge, init\ image)$$

$$edit\ image = \textbf{Image\ Edit}(init\ image, reflection\ advice)$$

### 3.4 QUESTION-ANSWER EVALUATION

We have borrowed the ideas from Paper2PosterPang et al. (2025) and proposed the "question answer" evaluation method. Specifically, we asked human experts (for textbook illustrations) and MLLM (for paper illustrations) to propose five multiple-choice questions for each reference image based on its content. Each multiple-choice question has five options, with A, B, C, and D being one correct option and three distractors. Option E is: Information is missing, unable to answer. The purpose of setting this option is to avoid the model guessing the correct answer and to track the missing information in the generated results.

$$question = \textbf{Question}(ref\ image)$$

When evaluating, we asked MLLM to answer these five multiple-choice questions based on the generated results of the multi-agent system above.

$$answer = \textbf{Answer}(edited\ image, question)$$

## 4 EXPERIMENT AND ANALYSIS

### 4.1 EXPERIMENTAL SETTING

#### 4.1.1 EXPERIMENTAL DATA

We have prepared two different datasets for the paper illustration generation task and the textbook illustration generation task. For the task of paper illustration generation, we used natural data from

Table 1: Our model is compared with the state-of-the-art open-source image generation models (Qwen-Image) and closed source image generation models (GPT-4o-Image) in terms of the results of paper and textbook illustration generation tasks. Among them, we will present the results of textbook illustration generation according to five different disciplines for better analysis and presentation.

|                  | Paper | Physics | Chemistry | Biology | Geography | Medicine |
|------------------|-------|---------|-----------|---------|-----------|----------|
| **Qwen-Image ACC**   | 0     | 0       | 0         | 0       | 0         | 0        |
| **GPT-4o-image ACC** | 0.781 | 0.69    | 0.62      | 0.69    | 0.76      | 0.66     |
| **Ours ACC**         | **0.88** | **0.79** | **0.77** | **0.89** | **0.88** | **0.85** |
| **Qwen-Image NBR**   | 0     | 0       | 0         | 0       | 0         | 0        |
| **GPT-4o-image NBR** | 0.835 | 0.71    | 0.67      | 0.71    | 0.79      | 0.69     |
| **Ours NBR**         | **0.927** | **0.81** | **0.8** | **0.91** | **0.93** | **0.89** |

SridBenchChang et al. (2025), totaling 220 pairs (illustration-caption-section). For the task of generating textbook illustrations, we invited PhDs from various fields to propose 20 basic illustration drawing questions in physics, chemistry, biology, geography, and medicine, totaling 100 questions. At the same time, they will propose five multiple-choice questions based on each question that can be answered using information from qualified illustrations.

### 4.1.2 PRETRAINED MODELS USED

In modules **Task Routing**, **Caption Format**, **Prompt Format**, **Keyword Extract** and **Reflection**, we use the GPT-4o API. In modules **Initial Drawing** and **Image Edit**, we use the GPT-4o-image API. In modules **Question** and **Answer**, we use the GPT-5 API.

### 4.1.3 EVALUATION METRIC

Based on the answer situation of one-way multiple-choice questions, we define two evaluation indicators: accuracy rate (ACC), which is the proportion of correctly answered questions to the total number of questions; Non-blank rate(NBR), which refers to the proportion of the total number of questions that did not select option E. The significance of setting these two evaluation indicators is that ACC can measure the scientific and logical nature of image generation; NBR can reflect the completeness of image generation information. Because image generation models may generate erroneous information, but this is different from not considering this information at all. The specific calculation formulas for these two indicators are as follows:

$$\text{ACC} = \frac{Number\ of\ correct\ answers}{Total\ number\ of\ questions}$$

$$\text{NBR} = 1 - \frac{Number\ of\ unanswered\ questions}{Total\ number\ of\ questions}$$

### 4.2 COMPARATIVE EXPERIMENT

From Tab.1, we can see that our multi-agent system has made significant improvements in two major tasks compared to the most advanced open source image generation model and closed source image generation model currently available. At present, the most advanced open-source image generation models can almost be considered to lack the ability to generate academic illustrations. However, the most advanced closed source image generation models have a significant gap compared to our proposed multi-agent systems in terms of both scientific and comprehensive image generation. At the same time, we can also see that our proposed multi-agent systems have advantages in various disciplines.

We have presented the generated results of the model mentioned above in Fig3. After comparing with the reference image, it can be seen that our multi-agent system can generate illustrations that are more similar to the reference image. Meanwhile, our results have clearer and more accurate

information. MAIG has significant advantages in both paper illustration generation and textbook illustration generation tasks.

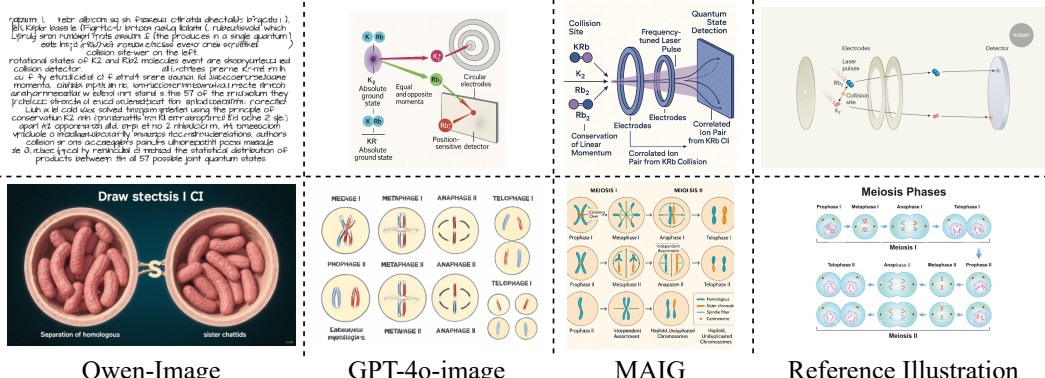

Qwen-Image      GPT-4o-image      MAIG      Reference Illustration

Figure 3: In the paper illustration generation task (above) and textbook illustration generation task (below), Qwen-Image, GPT-4o-image, and our multi-agent system(MAIG) compared the generated results with the reference image.

We can also see the difficulty of generating illustrations in different disciplines. As shown in Tab.1, the quality of image generation in biology and geography is higher than that in physics and chemistry. This indicates that current image generation models are better at generating visual content. The abstraction level of physics and chemistry illustrations is higher, which poses greater challenges for image generation models.

## 4.3 ABLATION STUDY

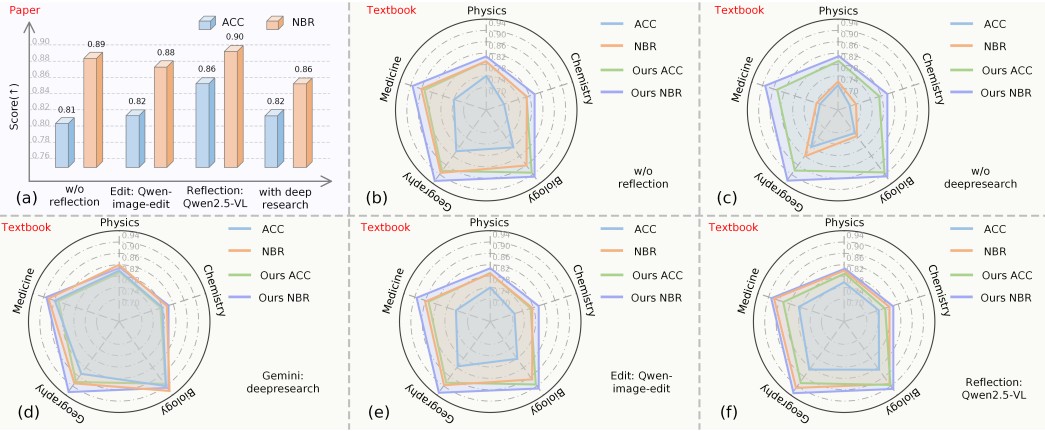

Figure 4: The ablation experiment results for our multi-agent system. (a) The effect of paper illustration generation in four scenarios (eliminating the reflection module, replacing the secondary editing model with Qwen Image edit, replacing the reflection model with Qwen2.5-VL 7B, and adding a deep research module); (b) After eliminating the reflection module, compare the generated results of textbook illustrations with the original results; (c) After eliminating the deep research module, compare the generated results of textbook illustrations with the original results; (d) Compare the generated results of textbook illustrations with the original results after replacing the deep research module with Gemini deep research; (e) After replacing the image editing module with Qwen-Image-edit, the generated results of the textbook illustrations were compared with the original results; (f) After replacing the reflection module with Qwen2.5-VL 7B, compare the generated results of the textbook illustrations with the original results.

### 4.3.1 REFLECTION MODULE

From Fig.4(a) and (b), we can see that the reflection module plays a crucial role in both paper illustration and textbook illustration. After removing the reflection and modification module, both ACC and NBR of the generated results showed a decrease. One noteworthy point is that the decrease in ACC is more pronounced compared to NBR, and the difference between ACC and NBR is also widening. This means that there are more error messages in the generated results. From this, it can be seen that the most important role of the reflection module is to correct erroneous information in the illustrations generated by beginners.

In fact, choosing the appropriate image editing model is equally important for the generation effect of the entire multi-agent system. From the Fig.4(a) and (e), we can see that when we replace the image editing module from GPT-4o-image to the most advanced open-source image editing model Qwen-Image-edit, the generation effect of the multi-agent system significantly decreases, whether in the task of generating paper illustrations or textbook illustrations.

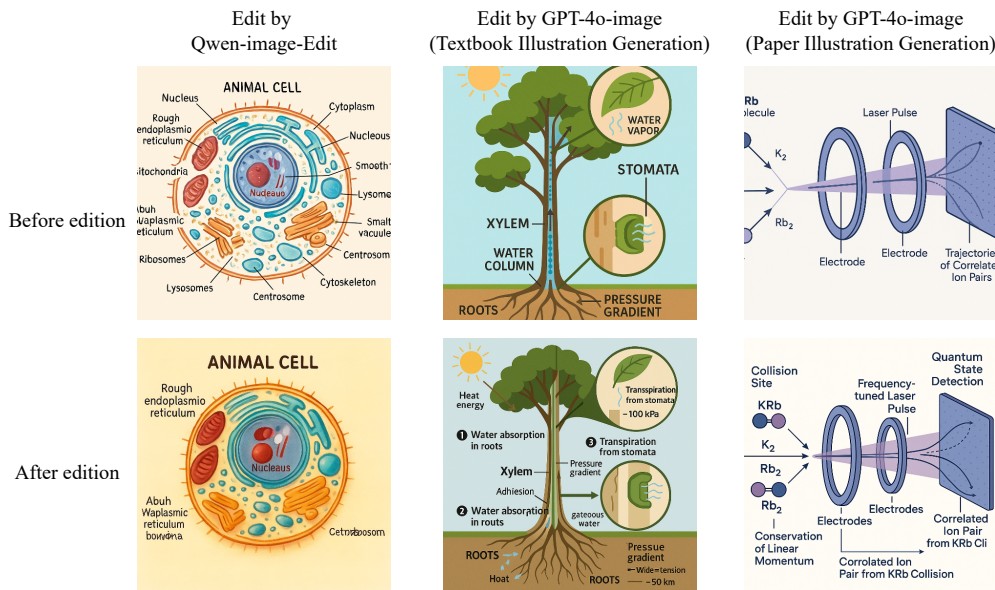

Figure 5: The impact of the editing module on the academic illustration generation effect of multi-agent systems. The first column compares the results before and after editing using Qwen-Image-Edit. The second column compares the results of textbook illustration generation before and after editing using GPT-4o-image. The third column compares the results of generating paper illustrations before and after editing with GPT-4o-image.

Fig.5 shows the impact of image editing models on the illustration results of multi-agent systems. It can be seen that some image editing models can even make the generated results worse. But using a high-performance image editing model will significantly improve the drawing quality. The information in the illustrations will become more extensive and the scientific rigor will also increase.

We also attempted to replace the reflection module with the open-source MLLM (Qwen2.5-VL 7B). From Fig.4(a) and (f), we can see that its effect is not as good as GPT-4o. This means that the type of reflective model has a significant impact on the effectiveness of multi-agent systems. After attempting to extract the reflective suggestions proposed by Qwen2.5-VL and GPT-4o, we found that GPT-4o can provide longer, richer, and more detailed suggestions.

### 4.3.2 DEEP RESEARCH MODULE

The role of Deep Research module has also been studied. From Fig.4(c), it can be seen that after removing the deep research module, the generation effect of textbook illustrations significantly decreases. Because the prompt words drawn in textbook illustrations often only contain the final requirements and lack a lot of key information. The role of Deep Research is to search for back-

ground information related to online search and illustration generation, which is crucial for textbook illustration generation tasks. It can be seen that both ACC and NBR have significantly decreased, indicating that the biggest problem after removing the deep research module is the lack of background information. Fig.6 shows that even without reflection and editing, deep research can still significantly improve the quality of illustration generation

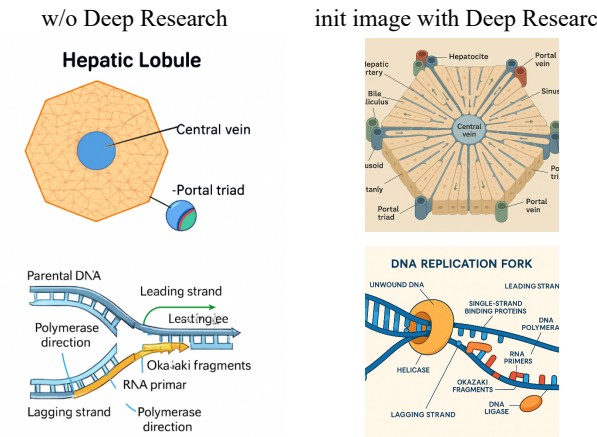

Figure 6: Comparison of illustration generation results using Deep Research and not using Deep Research under the same prompt words

As another well-known deep research model, Gemini deep research is used by us to replace GPT o3 deep research for comparison. From the Fig.4(d), it can be seen that compared to using GPT o3 deep research, Gemini deep research module performs approximately or even better on certain indicators. This indicates that in addition to GPT o3 deep research that we use, other high-performance deep research modules can also have a beneficial impact on multi-agent systems.

An interesting phenomenon is that when we add a deep research module in the process of paper illustration generation (i.e. extracting keywords and supplementing in-depth research information from prompt words organized using captions and sections), the effectiveness of multi-agent systems actually decreases, as shown in Fig.4(a). This indicates that targeted information (i.e. captions and sections provided by the authors in the paper) is often more effective in the task of paper illustration generation. The introduction of additional information can interfere with the generation of illustrations and is not conducive to their creation. The reason is that the task of generating illustrations for papers is a targeted and innovative task, which only requires innovative information provided by the author, rather than background knowledge from online searches (in fact, due to the innovation of the content, the knowledge searched may even be unrelated to the illustrations).

## 5 CONCLUSION AND DISCUSSION

In this paper, we propose MAIG: a multi-agent system for academic illustration generation. Using task routing, we divide the academic illustration generation task into two tasks: paper illustration generation and textbook illustration generation. We utilized in-depth research models, paper generation models, reflection models, and image editing models to draw academic illustrations, in order to improve the scientific accuracy and information completeness of the drawing results. At the same time, we proposed a model for evaluating the generated results using question answering, and proposed two indicators, ACC and NBR, to objectively evaluate the correctness and completeness of the generated results. Numerous experimental results have shown that our multi-agent system outperforms existing image generation models in academic illustration generation across different disciplines and tasks, achieving both scientific accuracy and information completeness. The ablation experiment proved the effectiveness and necessity of our proposed module in this task.

**Reproducibility Statement.** In order to facilitate further validation and research by researchers, we will make every effort to ensure that our research work is fully reproducible. We will publicly disclose the completed code repository and usage method of MAIG on Github in the future. Meanwhile, the datasets involved in our experimental section have already been made public or are about to be made public, including these two tasks. Our method relies on publicly available models, and all prompt words and specific model calling methods are detailed to ensure that the results are replicable.

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

## A   STATEMENT ON LLMS USAGE

The authors used large language models (LLMs) during the writing process solely for language refinement and editing. It should be explicitly stated that LLMs were not employed in any core aspects of the research, including the formulation of research ideas, the design of methodologies, the execution of experiments, or the development of conclusions. All scholarly contributions were made independently by the authors.

