# OpenReview forum: "MAIG: Multi-agent system for Academic Illustration Generation based on deep search and reflection"
_ICLR.cc/2026/Conference — ICLR 2026 Conference Desk Rejected Submission_

### Official Review · Reviewer_xRDB · 2025-10-15

**Soundness:** 2
**Presentation:** 2
**Contribution:** 1
**Rating:** 2
**Confidence:** 4

**Summary:**

MAIG integrates multiple agents to generate academic illustrations from textual descriptions. The authors identify key shortcomings in existing text-to-image models when applied to scientific domains, namely a lack of factual accuracy, informational completeness, and the inability to generate complex, unstructured conceptual visuals. The proposed MAIG system leverages a team of specialized AI agents, including a manager, a search agent, an illustration generation agent, and a reflection agent, to address these challenges. The workflow involves the search agent gathering relevant information, the generation agent creating an initial illustration using existing tools, and the reflection agent iteratively refining the output for accuracy and completeness. The authors use a QA-based evaluation metric to assess the quality of generated illustrations.

**Strengths:**

- The system is well-designed, e.g., the integration of external tools (Search, ECharts, Stable Diffusion) within the agentic workflow is a practical and effective method.

- The paper is easy to follow.

**Weaknesses:**

- It seems like an overclaim that "we also propose a novel Question-Answering (QA) based Evaluator". QA based evaluation is not a new thing.

- It is more like an engineering work rather than research. The overall concept of **using a multi-agent system for a complex task** has been explored extensively. The authors should, for instance, discuss how their framework compares to general-purpose multi-agent frameworks like AutoGen or CrewAI, and why a bespoke system was necessary.

- Insufficient comparison to recent agentic frameworks. The paper positions itself as a novel multi-agent system but fails to adequately discuss or compare its approach with a growing body of work on agentic workflows and multi-agent collaboration.

- The paper presents a very positive view of the MAIG system. A more critical discussion of its limitations and failure modes would be valuable. When does the system fail? Are there certain types of diagrams or concepts that it struggles with? How robust is the system to noisy or ambiguous inputs?

- typos, e.g., ”question (line150)

**Questions:**

- Do not want to mention this in the weakness that this paper is an okay application paper, stacking models and tools, but not sure if it is really related to "learning representations".

---

### Official Review · Reviewer_zpkA · 2025-10-30

**Soundness:** 3
**Presentation:** 3
**Contribution:** 2
**Rating:** 2
**Confidence:** 4

**Summary:**

The authors introduce an inference pipeline combining LLMs and image generation/editing models to automatically generate academic illustrations as raster graphics. The authors demonstrate that their approach outperforms other end-to-end methods through an automatic evaluation based on Q&A with VLMs.

**Strengths:**

* The authors tackle a challenging problem and achieve state-of-the-art results.
* The paper is well-written and easy to follow.

**Weaknesses:**

* The authors promote their Q&A-based evaluation framework as one of the core contributions, but later acknowledge that they have adopted this approach from prior work (l.248), which limits the novelty of their contribution.
* The other core contribution is the inference pipeline itself. While it performs well, it is essentially a wrapper around proprietary APIs, and the finding that investing more compute leads to better performance is hardly surprising.

**Questions:**

Text rendering appears to be a crucial component, do the authors evaluate this aspect? If not, how well did it perform in their experience?

---

### Official Review · Reviewer_26nX · 2025-10-30

**Soundness:** 2
**Presentation:** 2
**Contribution:** 2
**Rating:** 2
**Confidence:** 5

**Summary:**

Multi-agent pipeline (research → generate → reflect/edit) for scientific illustration; introduces an MLLM QA-based metric (ACC/NBR) for factuality/completeness; reports gains over GPT-4o/Qwen on small bespoke tasks.

**Strengths:**

- Clear problem framing (accuracy + completeness in academic figures).
- End-to-end system with module ablations.
- Tries to go beyond style metrics via automatic evaluation.

**Weaknesses:**

- **Thin novelty**: routing + research–reflect–edit mirrors prior multi-agent T2I/diagram pipelines, as well as QA-evaluator = MLLM-as-grader.
- **Questionable evaluation**: self-referential grading, ad-hoc ACC/NBR, small author-curated datasets.
- **Baselines dubious**: Qwen near-zero, unclear prompts/settings; no significance tests/error bars.
- **Reproducibility**: heavy reliance on closed models (GPT-4o/“GPT-5”).
- **Leakage/overfitting risk**: authors design questions and then evaluate with same paradigm.

**Questions:**

No questions

---

### Official Review · Reviewer_vR7j · 2025-10-30

**Soundness:** 1
**Presentation:** 3
**Contribution:** 2
**Rating:** 2
**Confidence:** 3

**Summary:**

The paper proposes two multi-stage generation pipelines for scientific Illustrations. If a context is available (as usual for scientific papers captions and texts), they (1) extract a description from the caption, (2) enrich it with details from the paper text, (3) generate an initial image, (4) reflect on the content and (5) edit the image, where every step is conducted by GPT. When context is scarce (like for generating a graphic for teaching scenarios), they first acquire context via web search. For analysis, they source 100 basic drawing questions for 5 scientific fields from PhD students and 220 caption-image pairs from SridBench. The 100 questions are proposed together with 5 multiple choice questions that can be answered from correct images. For the 220 images, the questions and answer options are generated by GPT5. Then, GPT5 is asked to answer the multiple choice questions based on the generated images. The ratio of correctly answered questions is reported as accuracy. Another metric is the ratio of images, where at least on question could be answered. They find that their pipeline provides improvements over plain prompting and open-source models.

**Strengths:**

- This is a very interesting and useful research topic for our community, as the successful image generation could save researchers and authors a lot of time.

- The paper provides many details on how their reported metrics will change if components of their pipeline will drop or are exchanged.

**Weaknesses:**

- I see the main weakness of the paper in using a rather unverified metric: asking GPT5 if questions (in >60% generated by GPT5) can be answered correctly based on the image. There is no human analysis whether GPT5 is able to perform this task correctly. Further, it is unclear, how the questions are distributed into correct and incorrect answers and whether the questions could also be answered without the image. The nature of the questions is also unclear, because the paper does not provide any examples. If the questions instead are of the nature "does this image show X?", the problem lies in the ability of GPT5 to check fine-grained image details. For example, the MAIG images that the paper presents in Figure 3 contain much more text, which might coax GPT5 into saying that content is actually represented. For example, for Meiosis, while the phases mostly follow the reference image, the things that are happening to the chromosomes inside the cells are largely without sense. The tree in Figure 5 has a lot of stray text like "water absorption through roots" at the top of the stem. DNA Ligase in Figure 6 is shown outside of the strand. In other words, while the images perhaps seem more correct in terms of visual similarity, important small details are missed, which could cause issues if the images were to be used in teaching, etc.. I suggest to (1) perform a human evaluation of some images, to see if the human accuracy matches that given by GPT5 and (2) to explore how biased the model is towards text shown in the images.

- While there is no human evaluation, the QA-based metric is also not compared to other potential metrics, like VQAScore with GPT backend.

- On another note, ratings might be biased if images are generated and rated by GPT models.

- Generally, it seems a bit unsurprising that a pipeline that is using much more resources on checking itself performs better. It would be interesting to contrast this with the costs of running the pipeline.

- I don't see the use-case for generating textbook illustrations without any changes. Given the power of current search engines, it would be much easier, to just find a matching image (perhaps also with an LLM). Also, it is unclear, to which degree the LLM will just reproduce memorized graphics

- As you describe, there are many works that consider Scientific Figure Generation. Perhaps, it would be interesting to compare how your pipeline will perform on these.

**Small things:**
- In line 456, you suddenly write GPT-o3
- I would suggest to use a different format for the variable names, as the text appears quite spaced out

**Questions:**

- Tying in with the weaknesses, my question would be the same: Can you provide some examples of image - question pairs that were graded with GPT-5 as example?

- Also, do these questions have correct and incorrect options? If the latter is the case you should probably report the F1 score.

---

### Note · Program_Chairs · 2026-01-17
**Submission Desk Rejected by Program Chairs**

The following references in this submission do not refer to real documents and/or have major errors in bibliographic information:

 J. Qiu, Y. He, X. Juan, Y. Wang, Y. Liu, Z. Yao, Y. Wu, X. Jiang, and L. Yang. Mosaig: Multi-agent multimodal models for multicultural t2i. arXiv preprint arXiv:2502.14345, 2025c.
J. Chen, X. Hou, Y. Jiang, and Y. Wu. T2i-copilot: Multi-agent collaborative text-to-image generation. In Proceedings of the IEEE/CVF International Conference on Computer Vision (ICCV), 2025 .
J. Liu, Y. Zhang, X. Li, and Z. Wang. From image generation to infrastructure design: A multi-agent approach. arXiv preprint arXiv:2502.14345, 2025d.
J. Liu, Y. Zhang, X. Li, and Z. Wang. Comfybench: Benchmarking multi-agent workflows for image generation. In Proceedings of the IEEE/CVF Conference on Computer Vision and Pattern Recognition (CVPR), 2025a.